# Fast Reliability Estimation for Neural Networks with Adversarial Attack-Driven Importance Sampling

**Karim Tit**[1,2]  **Teddy Furon**[2]

[1]University of Luxembourg, Luxembourg, LU*
[2]Inria, CNRS, IRISA, University of Rennes, Rennes, FR

## Abstract

This paper introduces a novel approach to evaluate the reliability of Neural Networks (NNs) by integrating adversarial attacks with Importance Sampling (IS), enhancing the assessment's precision and efficiency. Leveraging adversarial attacks to guide IS, our method efficiently identifies vulnerable input regions, offering a more directed alternative to traditional Monte Carlo methods. While comparing our approach with classical reliability techniques like FORM and SORM, and with classical rare event simulation methods such as Cross-Entropy IS, we acknowledge its reliance on the effectiveness of adversarial attacks and its inability to handle very high-dimensional data such as ImageNet. Despite these challenges, our comprehensive empirical validations on the datasets the MNIST and CIFAR10 demonstrate the method's capability to accurately estimate NN reliability for a variety of models. Our research not only presents an innovative strategy for reliability assessment in NNs but also sets the stage for further work exploiting the connection between adversarial robustness and the field of statistical reliability engineering.

## 1 INTRODUCTION

In the fast-evolving landscape of Deep Learning, ensuring the robustness and reliability of Neural Networks (NNs) is paramount, particularly for critical decision-making applications. This work introduces a simple approach for estimating the local robustness of trained Neural Networks against uncertainties, with a focus on their performance in the vicinity of clean inputs. We propose a method that combines adversarial attacks with the Importance Sampling (IS) technique.

Adversarial attacks, traditionally aimed at uncovering NN vulnerabilities, are repurposed in our methodology as a strategic guide for the IS process. The point of this approach is to identify the most error-prone regions in the input space, thus directing the sampling process contrary to the commonly used Crude Monte Carlo method.

A key contribution of this research is the comparative analysis of our method with classical techniques from the field of Statistical Reliability Engineering [Der Kiureghian, 2022]. These techniques include the First Order Reliability Method (FORM), Second Order Reliability Method (SORM), and Line Sampling [Koutsourelakis et al., 2004], which have not been extensively applied to DNNs in very high-dimensional spaces, a gap our study aims to fill.

In addition, we compare this IS estimator to classical rare event simulation algorithms. These include Cross-entropy-based Adaptive Importance Sampling (CE-AIS) [Rubinstein and Kroese, 2016] and Adaptive Multilevel Splitting (AMS) [Au and Beck, 2001] methods. We show that the proposed method is more efficient and faster than these techniques for various architectures and datasets.

However, this novel estimator is not without limitations. Its effectiveness is inherently tied to the efficiency of adversarial attacks; it can only be as good as the adversarial attacks it relies on. Moreover, the occurrence of weight degeneracy in extremely high-dimensional data, such as ImageNet data where $d = 150528$, restricts the applicability of this method. These constraints highlight the need for a continuum of solutions from fast methods, like the one proposed here, to more advanced but slower methods for complex settings.

This paper delves into the intricacies of integrating adversarial attack strategies within the IS framework, addressing both the algorithmic challenges and the theoretical aspects. We focus on adapting these strategies for high-dimensional reliability analysis in NNs, confronting computational and conceptual hurdles. We validate our approach through empirical studies and experiments on a variety of deep learning models using the computer vision datasets MNIST and

---

*The majority of this work was carried out while Karim Tit was a Ph.D. candidate at the University of Rennes.

*Accepted for the 40th Conference on Uncertainty in Artificial Intelligence* (UAI 2024).

CIFAR10. These evaluations demonstrate the method's efficacy in rapidly estimating NN probabilistic robustness.

## 2 PROBLEM STATEMENT AND RELATED WORK

Certified robustness refers to the ability of a neural network to consistently classify inputs correctly within a specified range of perturbations. Unlike empirical robustness, which is tested through experiments and simulations, certified robustness provides theoretical assurances, ensuring that the network's predictions remain unchanged for perturbations below a certain magnitude. Various approaches have been developed to certify the robustness of neural networks.

*Complete Verification* provides formal guarantees of robustness by exhaustively analyzing all possible perturbations within a given range. Katz et al. [2017] proposed the first exact verification method for Neural Networks, using tools from Satisfiability Modulo Theories (SMT). Notably, they prove in the same work that this is an NP-complete problem.

*Incomplete Verification Methods* use conservative approximations. They are computationally efficient but incomplete. Interval bound propagation and abstract interpretation are prominent examples Singh et al. [2018].

*Probabilistic Assessment* resorts to random simulations with statistical guarantees on the probability of failure under a certain noise distribution Webb et al. [2019]. Some combines these with formal methods Weng et al. [2019]. Our method pertains to this family.

### 2.1 PROBABILISTIC ASSESSMENT

Consider a trained neural network classifier $f : [0,1]^d \to [0,1]^C$ mapping an input to a probability vector for $C$ classes and a clean input $\mathbf{x}_0$ which is well classified: $\arg\max_{1 \leq i \leq C} f_i(\mathbf{x}_0) = c$, where $c$ is the ground truth class. The question is whether a random perturbation, modeling uncertainties on the input measurement, can cause a misclassification.

The approach of Webb et al. [2019] is to cast this issue as a probability measure. Assuming a statistical model of a random additive perturbation $\mathbf{N}$, the objective is to compute the probability of failure (i.e. misclassicification). We introduce the random input $\mathbf{X} = \mathbf{x}_0 + \mathbf{N}$ whose distribution is denoted $\pi$. The probability of a failure is defined as

$$P_{\mathrm{F}}(\pi) := \int_{[0,1]^d} \mathbb{1}\left[h(\mathbf{x}) \geq 0\right] \pi(d\mathbf{x}), \qquad (1)$$

where $h : [0,1]^d \to [-1,1]$ computes how close an input is from a misclassification. For instance,

$$h(\mathbf{x}) := \max_{i \in [1:C], i \neq c} f_i(\mathbf{x}) - f_c(\mathbf{x}). \qquad (2)$$

$h(\mathbf{x}) > 0$ indicates that $\mathbf{x}$ is not classified as class $c$, the ground truth of $\mathbf{x}_0$.

### 2.2 RELATED WORKS

Recent machine learning papers dealing with local robustness against uncertainties ignore the literature of Statistical Reliability Engineering and refer more to works in the field of Rare Event Simulation. The workhorse is mainly the Sequential Monte Carlo (SMC) (also knows as Adaptive Multilevel Splitting (AMS)) family of algorithms [Au and Beck, 2001, Cérou et al., 2019].

As far as we know, Webb et al. [2019] are the first to use an SMC simulation to estimate the probability of failure of deep NNs. Tit et al. [2021] use a variant that is faster but only predicts whether the probability of failure is below a critical level. The method has some statistical guarantees and is efficient since the reported critical level can be as low as $10^{-50}$.

Baluta et al. [2021] use the Crude Monte Carlo simulation though within a sequential testing scheme [Wald, 1945], which increases the computational budget adaptively. It comes with robust non-asymptotical guarantees but in practice only works for high critical levels, typically greater than $10^{-3}$. These methods need the statistical model of the uncertainties, and also the function $h$ (2) (if working in the input space) or function $G$ (3) (if working in the U-space) as a black box.

Tit et al. [2023] propose a new SMC-like algorithm tailored for NNs: it exploits the gradient $\nabla G(\mathbf{u})$ which is easy to compute for *white box* NNs thanks to auto-differentation via backpropagation.

However, all these variants of SMC consume a lot of calls to the neural network function. Indeed, the total number of calls is generally on the order of *hundreds of thousands* for making a statement about the probability of failure around a *single* input $\mathbf{x}_0$. In contrast, our method, under the assumptions we detail, gives reliable estimations in a few thousand calls.

## 3 BACKGROUND

### 3.1 STATISTICAL RELIABILITY ENGINEERING

The problem stated in (1) is exactly the core issue in Statistical Reliability Engineering, a domain born in the 70s. Here, $h$ is a state function of a physical system described by parameters stored in $\mathbf{x}$. The system is reliable when $h(\mathbf{x}) \leq 0$, which is the case around the nominal state $\mathbf{x}_0$. The state $\mathbf{X}$ deviates from $\mathbf{x}_0$ due to some random uncertainties. The number of parameters is usually small and the state function has a close form inherited from the rules of physics.

However, the computation of (1) is difficult because $\pi$ or the region $\{h(\mathbf{x}) \geq 0\}$ is complicated.

### 3.1.1 Most Probable Failure Point in the U-space

To get an abstraction from the distribution $\pi$, one usually considers that there exists a bijective isoprobabilistic transformation $\mathcal{T}$ that pushes forward the normal distribution to $\pi$. In other words, $\mathbf{X} = \mathcal{T}(\mathbf{U}) \sim \pi$ when $\mathbf{U} \sim \mathcal{N}(\mathbf{0}; \mathbf{I})$. Examples are the Nataf [1962] and Rosenblatt [1952] transformations. This rephrases the problem into

$$P_{\mathrm{F}}(\pi) = \mathbb{E}[\mathbb{1}\left[G(\mathbf{U}) \leq 0\right]] \tag{3}$$

where $G := -h \circ \mathcal{T}^{-1}$.

The following methods approximate the failure event around the Most Probable Failure Point (MPFP), also called the design point. It is defined as the point in the U-space with the highest probability density on the frontier $G(\mathbf{u}) = 0$. Formally:

$$\mathbf{u}^* := \arg \max_{\mathbf{u}:G(\mathbf{u})=0} \phi(u) = \arg \min_{\mathbf{u}:G(\mathbf{u})=0} \|\mathbf{u}\|^2. \tag{4}$$

In classical applications of Statistical Reliability Engineering, finding this point is usually not difficult because it has a closed form or a numerical solution like the HL-RF algorithm (Hasofer and Lind [1974], Rackwitz and Flessler [1978]) quickly converges in a low dimensional space. Sect. 5 shows this is still possible on small-scale images.

### 3.1.2 FORM and SORM

The First (resp. Second) Order Reliability Method FORM (resp. SORM) models $G(\mathbf{u})$ by a linear (resp. quadratic) function in the neighborhood of $\mathbf{u}^*$. This leads to the following approximations:

$$P_{\mathrm{F}}^{\mathrm{FORM}} := \Phi(-\|\mathbf{u}^*\|_2), \tag{5}$$

$$P_{\mathrm{F}}^{\mathrm{SORM}} := \Phi(-\|\mathbf{u}^*\|_2) \prod_{i=1}^{d-1}(1 + \kappa_i)^{-1/2}. \tag{6}$$

where $(\kappa_i)_{i=1}^{d-1}$ are the eigenvalues of the Hessian matrix of $G$ at point $\mathbf{u}^*$ restricted to the subspace orthogonal to $\mathbf{u}^*$, denoted $\mathsf{span}(\mathbf{u}^*)^{\perp}$. The product accounts for the curvatures of the frontier around $\mathbf{u}^*$, thereby refining the probability of failure estimate compared to FORM. We illustrate this phenomenon in section 5, for small-scale images, as it is not possible to apply form to larger images due to its computational complexity in $O(d^2)$.

### 3.1.3 Line Sampling (LS)

LS also accounts for curvature, though, without using the Hessian matrix [Koutsourelakis et al., 2004]. It is a ran-

dom simulation that has advantages for complex and high-dimensional systems. In a nutshell, it draws random normal vectors $\mathbf{U}_i$, projects them onto hyperplane $\mathcal{H} = \mathsf{span}(\mathbf{u}^*)^{\perp}$ and finds the minimum $\beta_i$ s.t. $G(\mathbf{U}_i^{\perp} + \beta_i \mathbf{u}^*/\|\mathbf{u}^*\|) = 0$. See also Figure 1. The final estimator is given by:

$$P_{\mathrm{F}}^{\mathrm{LS}} := \frac{1}{N}\sum_{i=1}^{N}\Phi(-\beta_i). \tag{7}$$

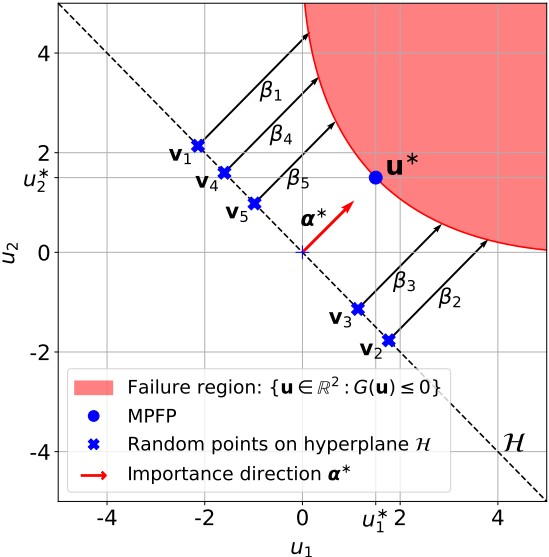

Figure 1: Illustration of Line Sampling in $\mathbb{R}^2$.

### 3.1.4 Importance Sampling (IS)

The methods above assume that the design point $\mathbf{u}^*$ is easily computed. Without this assumption, the Crude Monte Carlo estimator

$$P_{\mathrm{F}}^{\mathrm{CMC}} := \frac{1}{N}\sum_{i=1}^{N}\mathbb{1}\left[G(\mathbf{U}_i) \leq 0\right] \tag{8}$$

is a possibility only if the true probability $P_{\mathrm{F}}$ is not small because the relative estimator variance scales as $1/N P_{\mathrm{F}}$.

Importance Sampling is an alternative estimator:

$$P_{\mathrm{F}}^{\mathrm{IS}} := \frac{1}{N}\sum_{i=1}^{N}\mathbb{1}\left[G(\mathbf{Y}_i) \leq 0\right]\frac{\phi(\mathbf{Y}_i)}{f_Y(\mathbf{Y}_i)}, \tag{9}$$

where $\mathbf{Y}_i$ are i.i.d. random vectors whose p.d.f. is denoted $f_Y$ and $\phi$ is the p.d.f. of the standard normal law. It may bring a variance reduction if the p.d.f. of $\mathbf{Y}$ is similar to the optimal $f_Y^{\star}(\mathbf{Y}) \propto \phi(\mathbf{Y})\mathbb{1}\left[G(\mathbf{Y}) \leq 0\right]$.

Without any prior knowledge about $G$, it is difficult to figure out where the region $\{\mathbf{U}|G(\mathbf{U}) \leq 0\}$ is located in the U-space, hence the shape of the optimal density $f_Y^{\star}$. The

Cross-Entropy method makes a progressive exploration of the space by iteratively sampling random vectors of density $f_Y^{(j)}$ and exploit the variables $\left( \mathbb{1} \left[ G \left( \mathbf{Y}_i^{(j)} \right) \leq \tau_j \right] \right)_i$ to refine the density $f_Y^{(j+1)}$ and the threshold $\tau_{j+1}$ for the next iteration [Rubinstein and Kroese, 2016, Chap. 8]. For instance, if we restrict to the un-centered normal laws family $\mathcal{N}(\boldsymbol{\theta}, \mathbf{I})$, then $f_Y^{(j+1)}$ is characterized by its mean value

$$\boldsymbol{\theta}^{(j+1)} := \frac{\sum_{i=1}^{N} \mathbb{1} \left[ G \left( \mathbf{Y}_i^{(j)} \right) \leq \tau_j \right] \frac{\phi(\mathbf{Y}_i)}{f_Y(\mathbf{Y}_i)} \mathbf{Y}_i^{(j)}}{\sum_{i=1}^{N} \mathbb{1} \left[ G \left( \mathbf{Y}_i^{(j)} \right) \leq \tau_j \right] \frac{\phi(\mathbf{Y}_i)}{f_Y(\mathbf{Y}_i)}}. \quad (10)$$

## 3.2 ADVERSARIAL EXAMPLES

Adversarial examples are considered a vulnerability of machine learning classifiers. Given an input $\mathbf{x}_0$ well classified by classifier $c(\cdot)$, the adversarial example is the nearest misclassified input:

$$\mathbf{x}^\star = \arg \min_{\mathbf{x} \in [0,1]^d : c(\mathbf{x}) \neq c(\mathbf{x}_0)} d(\mathbf{x}, \mathbf{x}_0), \quad (11)$$

where $d(\mathbf{x}, \mathbf{x}_0)$ is a distance between $\mathbf{x}$ and $\mathbf{x}_0$. For the case where the classifier is a neural network, the event $c(\mathbf{x}) \neq c(\mathbf{x}_0)$ can be rephrased as $h(\mathbf{x}) \geq 0$ (see (2)).

If distance $d$ is the Euclidean norm of $\mathbf{x} - \mathbf{x}_0$, then the adversarial example (11) *in the U-space* is indeed the design point (4). As far as we know, this connection between adversarial examples and statistical reliability engineering has never been made before. This implies that algorithms from this later domain, like HL-RF designed in the 70s, could find $\ell_2$ adversarial examples. This is indeed not the case due to the high dimensionality of the input space in modern classification problems. The recent attacks finding adversarial examples are more efficient.

The Carlini and Wagner [2017] (CW) attack is known for its precision in scouting adversarial examples with minimal perturbation. It amounts to solve the Lagrangian formulation of (4): Define $J(\mathbf{u}, \lambda) := \|\mathbf{u}\|^2 + \lambda G(\mathbf{u}), \forall \lambda \leq 0$ and

$$\mathbf{u}_\lambda^\star := \arg \min_{\mathbf{x} \in [0,1]^d} J(\mathbf{u}, \lambda). \quad (12)$$

This is done with a numerical solver. On top of it, a line search finds $\lambda^\star$ s.t. $G(\mathbf{u}_{\lambda^\star}^\star) = 0$. This attack requires a fair amount of function $G$ gradient computations. Of note, we have the following property: $2\mathbf{u}_\lambda^\star + \lambda \nabla G(\mathbf{u}_\lambda^\star) = \mathbf{0}$, or:

$$\cos \left( \mathbf{u}_\lambda^\star, \nabla G(\mathbf{u}_\lambda^\star) \right) = -1. \quad (13)$$

The FMNA attack [Pintor et al., 2021] (abbreviation for "Fast Minimum-norm Adversarial Attack"), focuses on finding the shortest path to the decision boundary, iteratively refining the input to project it onto the decision boundary. This method is much faster and almost as precise as CW.

## 4 PROPOSED METHOD

This paper introduces a simple yet innovative approach to speed up the reliability estimation of Neural Networks by integrating adversarial attacks into the framework of Importance Sampling (IS). This method is built upon the foundations of Statistical Reliability Engineering and especially MPFP-based Importance Sampling [Melchers and Beck, 2018]. It leverages the strengths of specific adversarial attacks to construct a biased distribution for more effective sampling. The key lies in using these attacks to shift the focus of the sampling process towards regions in the input space where the NN is most vulnerable, thus allowing for a more accurate estimation of the model's reliability.

### 4.1 CONSTRUCTING THE BIASED DISTRIBUTION

Utilizing these adversarial attacks, we construct a shifted Gaussian distribution in the U-space (standard normal space), where the mean of the distribution is adjusted based on the insights gained from the attack. This results in a biased distribution that is centered around the region of high failure probability. The steps for constructing this distribution are as follows:

**Mapping to the U-Space:** Transform the evaluation of (1) into the estimation of (3) as explained in Sect. 3.1.1. In the U-space, the uncertainties are standard normally distributed.

**Generating Adversarial Examples:** Employ attacks described in Sect. 3.2 to find the adversarial example $\mathbf{u}^\star$ that highlights the NN's vulnerable point. Select an attack efficient in high-dimensional spaces and designed to find adversarial examples of *minimal* norm, like CW or FMNA.

**Creating the Biased Distribution:** Formulate a Gaussian distribution in the U-space centered around the adversarial example, ensuring that the sampling process is concentrated around the most vulnerable regions of the NN. Run the Importance Sampling procedure with $\mathbf{Y}_i \overset{i.i.d.}{\sim} \mathcal{N}(\mathbf{u}^\star, \mathbf{I})$. This means that the ratio appearing in (9) equals

$$\frac{\phi(\mathbf{Y}_i)}{f_Y(\mathbf{Y}_i)} = \exp \left( \|\mathbf{u}^\star\|^2/2 - \mathbf{Y}_i^\top \mathbf{u}^\star \right). \quad (14)$$

### 4.2 ASSUMPTIONS

This method relies on the following assumptions:

**A1.** The design point is unique. This means that $\mathbf{u}^\star$ is a global minimum of $J(\mathbf{u}, \lambda^\star)$. If existing, local minima lie further away from the origin. This means that the probability of failure is dominated by the probability of sampling $\mathbf{U}$ around this unique design point s.t. $G(\mathbf{U}) > 0$.

**A2.** The attack finds this design point.

**A3.** The frontier locally around the design point $\mathbf{u}^\star$ is not so curved.

Once the attack produces a point $\mathbf{u}^\star$, it is easy to check that it lies on the boundary, i.e. $G(\mathbf{u}^\star) = 0$, and it is a local minimum because (13) holds. However, this does not prove that $\mathbf{u}^\star$ is the true global minimum. As for assumption A3, if too many random vectors $\mathbf{Y}_i$ drawn for the IS lead to $G(\mathbf{Y}_i) \geq 0$, it means that the Importance Sampling estimation (9) will be zero or dominated by too few samples. Statisticians say that the *efficient* number of samples is too small which provokes a non-reliable estimation. In conclusion, we have means for controlling that assumption A3 holds and assumption A2 is partly fulfilled. Yet, it is impossible to ensure that A1 holds.

# 5 EXPERIMENTAL RESULTS

## 5.1 EXPERIMENTAL SETUP

We compare the convergence of different Rare Event Simulation methods: our Adversarial-Attack Driven IS of Sect. 4 (which we abbreviate by ADV-IS), the Line Sampling (LS) estimator (7), the Cross-Entropy Importance Sampling (CE-IS) (9) (10), and two estimators based on Sequential Monte Carlo (SMC) techniques, the Multilevel Splitting [Au and Beck, 2001] and a Langevin Monte Carlo within an SMC scheme [Tit et al., 2023], that we note respectively MLS-SMC and MALA-SMC (MALA stands for Metropolized Langevin Algorithm). An important parameter for these SMC methods, in addition to the number of samples $N$, is the number $T$ of applications of a transition kernel, which reduces the dependence between samples. Theoretical guarantees are derived under the perfect independence ($T = \infty$). In practice, $T < \infty$ has a huge impact on the number of calls to the NN.

We consider three models across two datasets and apply uniform noise to different instances. For each instance, we compute a reference probability of failure $\hat{P}_F^{\text{Ref}}$ by using an expensive IS compute (taking $N$ of the order $10^6$) and we check a posteriori that all methods converge towards the same value. In addition to benchmarking the rare event simulation methods, we compute both the FORM estimate $P_F^{\text{FORM}}$ and, whenever possible, the SORM estimate $P_F^{\text{SORM}}$, as defined above, using different search methods. These estimators are quantitively compared thanks to two metrics:

- The coefficient of variation $\Delta[\cdot]$, defined for an estimator $\hat{P}_F$ as, $\Delta[\hat{P}_F] = \frac{\sqrt{\mathbb{V}[\hat{P}_F]}}{\mathbb{E}[\hat{P}_F]}$.
- The relative mean absolute error, note $\text{RE}[\cdot]$, define as: $\text{RE}[\hat{P}_F] = \mathbb{E}[|P_F - \hat{P}_F|] \cdot P_F^{-1}$.

In practice, we have to estimate these metrics by their empirical counterpart. Moreover, as RE explicitly involves the

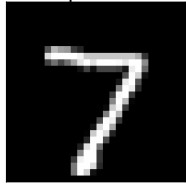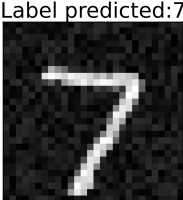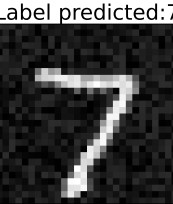

Figure 2: Input $\mathbf{x}_{0,1}$ (on the left) and examples of perturbations with uniform noise $\varepsilon = 0.18$.

failure probability, we will use the reference probability $\hat{P}_F^{\text{Ref}}$ as a surrogate. Crucially, for a fair comparison, these metrics and the complexity of an estimator (gauged by the number of calls) are measured over the same runs. All experiments were run on a personal laptop, with a 4060RTX GPU. All the code will be made available publicly on GitHub once the reviewing will be over.

## 5.2 MNIST

### 5.2.1 MLP with two hidden layers

We first compare these methods via experiments on a simple Multi-Layer Perceptron (MLP) with only 2 hidden layers (each containing 200 neurons) trained on the MNIST dataset [LeCun et al., 1990], which will be referred to as model $\mathsf{M}_1$, and on a first instance we note $\mathbf{x}_{0,1}$. We consider an additive noise perturbation, uniform on the $\ell_\infty$ ball of radius $\varepsilon = 0.18$ and centered on $\mathbf{x}_{0,1}$, see Figure 4. This distribution can be mapped to the standard Gaussian law via the isoprobabilistic transform mentioned in Sect. 3.1.1. At this level of noise, the probability of misclassification is low. Running an expensive simulation we find that $\hat{P}_F^{\text{Ref}} \approx 1.95 \cdot 10^{-6}$.

We apply the FORM and SORM methods with three adversarial attacks, the Carlini-Wagner attack, FMNA attack, and HLRF attacks. Indeed, the dimension is $d = 784$ for this dataset and it is possible to manipulate matrices of size $d \times d$ and in particular to evaluate, via auto-differentiation, the Hessian of $G$. Table 1 presents the results. At a glance, it is clear that FORM significantly overestimates the probability of failure when the FMNA and HLRF attacks find the design point (4), but underestimates it with the CW attack. This indicates that the decision boundary at $u^\star$ is not "flat" enough for a linear approximation to hold. This idea is further reinforced by observing that the SORM estimators are indeed closer to the actual probability of failure. In addition, we note that, here, the CW attack performed poorly, as its norm is higher in comparison with that of the two other attacks. Moreover, the Hessian $\nabla^2 h = -\nabla^2 G$ has both positive and negative eigenvalues at the CW point, whereas it only has non-positive eigenvalues at the other attack points.

We next, look at the convergence of the statistical methods

Table 1: FORM/SORM estimations of $\hat{P}_\mathrm{F}^\mathrm{Ref} \approx 1.95 \cdot 10^{-6}$ for model $\mathsf{M}_1$ and input $\mathbf{x}_{0,1}$, with uniform noise ($\varepsilon = 0.18$).

| Attack | $P_\mathrm{F}^\mathrm{FORM}$ | $P_\mathrm{F}^\mathrm{SORM}$ | $\cos(\tilde{u}^*, \nabla G(\tilde{u}^*))$ |
|---|---|---|---|
| CW | $7.2 \cdot 10^{-8}$ | $6.39 \cdot 10^{-6}$ | $-0.69$ |
| FMNA | $1.17 \cdot 10^{-4}$ | $6.49 \cdot 10^{-6}$ | $-0.995$ |
| HLRF | $7.53 \cdot 10^{-5}$ | $6.65 \cdot 10^{-6}$ | $-0.977$ |
| | $\|\tilde{u}^*\|_2$ | $G(\tilde{u}^*)$ | Time (in sec.) |
| CW | $5.26$ | $-4.1 \cdot 10^{-5}$ | $0.19$ |
| FMNA | $3.68$ | $-1.4 \cdot 10^{-5}$ | $0.16$ |
| HLRF | $3.79$ | $-2.0 \cdot 10^{-2}$ | $0.01$ |

with respect to the average number of calls, noted $\bar{N}_\mathrm{calls}$. In Figure 5 we see that all methods seem to converge towards the reference probability as the average number of calls increases, though their convergence rate differs. In particular, the Sequential Monte Carlo methods, MALA-SMC and MLS-SMC, converge noticeably slower than the LS and ADV-IS methods. The cross-entropy (CE) IS method has a significant overhead as it must first converge towards a good parameter $\boldsymbol{\theta}$, before exploiting its final distribution to compute an estimate of $P_\mathrm{F}$. We focus on the IS and LS methods in Figure 6, comparing their speed of convergence for different adversarial attacks. These figures are obtained by: running each method 400 times (with different random seeds to obtain standard errors) using a given number of samples $N$ and repeating the same operation for increasing values of $N$. For example, we ran the ADV-IS for values of $N$ in the range $\{100, 1000, 10000, 50000, 100000\}$.

Finally, we give the best performance of each algorithm (with respect to the number of samples used) in terms of the coefficient of variation multiplied by a measure of the

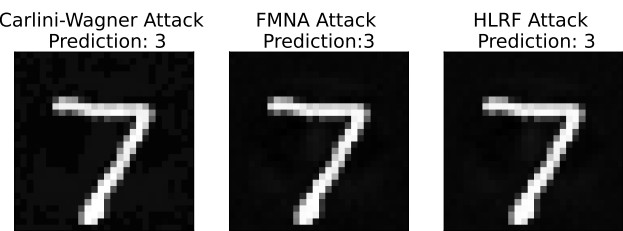

Carlini-Wagner Attack
Prediction: 3

FMNA Attack
Prediction: 3

HLRF Attack
Prediction: 3

Figure 3: Adversarial attacks for model $\mathsf{M}_1$ on input $\mathbf{x}_{0,1}$.

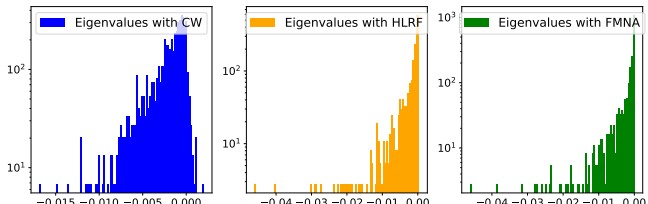

Figure 4: Eigenvalues of the Hessian of $h$ at the CW attack (on the left), at the FMNA attack (in the center), and the HLRF attack (on the right).

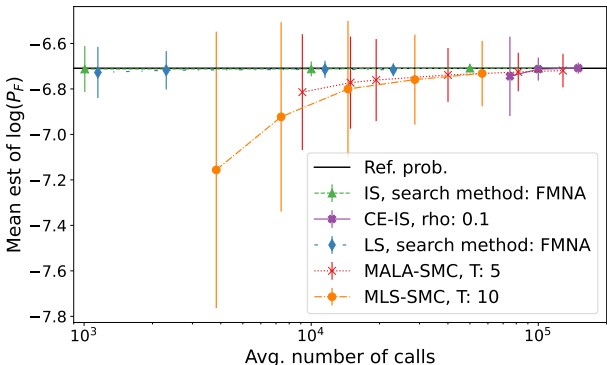

Figure 5: Convergence of different estimators w.r.t. the number of calls to the model $\mathsf{M}_1$.

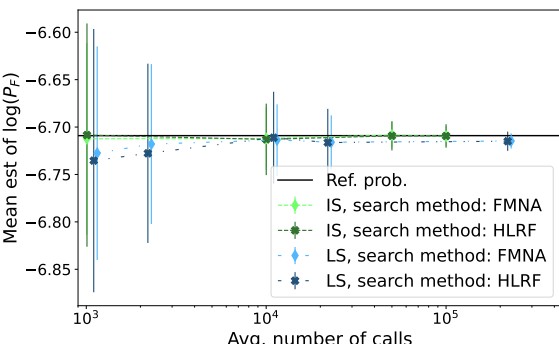

Figure 6: Convergence of IS and LS with different attacks.

computational burden. In practice, we use either the number of calls to the model $\bar{N}_\mathrm{calls}$ (i.e. the metric $\hat{\Delta}^2[\hat{P}_\mathrm{F}] \times \bar{N}_\mathrm{calls}$), or the duration of the simulation in seconds (i.e. the metric $\hat{\Delta}^2[\hat{P}_\mathrm{F}] \times$ time). Table 2 reports the results where $N_\mathrm{best}$ denotes the number of samples that gave the best performance in terms of the metric $\hat{\Delta}^2[\hat{P}_\mathrm{F}] \times \bar{N}_\mathrm{calls}$. All metrics reported in this table pertain to the ADV-IS method outperforms all other methods, for both metrics mentioned above. The CE-IS method also obtains good performance, for a relatively low number of samples $N_\mathrm{best}$ used for estimation. However, the *total* number of calls needed for CE-IS is in the order of *hundreds of thousands*.

### 5.2.2 MLP with four hidden layers

We now consider a similar MLP architecture with four hidden layers (each hidden layer containing 200 neurons), denoted $\mathsf{M}_2$. Simulation results for the FORM and SORM algorithms are given in the Appendix. Overall, these results support the idea that the decision boundaries of neural networks do not appear to be (locally) flat enough to be accurately approximated by hyperplanes, as the FORM method tends to overestimate the probability by an order of 10 or more. In contrast, the SORM method shows promising re-

Table 2: Best performance of estimators of $P_{\mathrm{F}}$ for the model $M_1$ and input $\mathbf{x}_{0,1}$, with uniform noise ($\varepsilon = 0.18$).

| Method | $N_{\mathrm{best}}$ | time (sec.) | $\mathrm{RE}[\hat{P}_{\mathrm{F}}]$ |
|---|---|---|---|
| ADV-IS | $5 \cdot 10^4$ | $5 \cdot 10^{-2}$ | $2.5 \cdot 10^{-2}$ |
| CE-IS | $3 \cdot 10^4$ | $2.3 \cdot 10^{-1}$ | $4.3 \cdot 10^{-2}$ |
| LS | $50$ | $4.3 \cdot 10^{-2}$ | $2.1 \cdot 10^{-1}$ |
| MALA | $256$ | $2.0 \cdot 10^{-1}$ | $2.1 \cdot 10^{-1}$ |
| MLS | $1024$ | $2.5 \cdot 10^{-2}$ | $2.6 \cdot 10^{-1}$ |
| | $\hat{\Delta}^2[\hat{P}_{\mathrm{F}}] \times \bar{N}_{\mathrm{calls}}$ | $\hat{\Delta}^2[\hat{P}_{\mathrm{F}}] \times$ time | $\bar{N}_{\mathrm{calls}}$ |
| ADV-IS | $48$ | $4.8 \cdot 10^{-5}$ | $5 \cdot 10^4$ |
| CE-IS | $460$ | $7 \cdot 10^{-4}$ | $1.5 \cdot 10^5$ |
| LS | $77$ | $2.9 \cdot 10^{-3}$ | $1200$ |
| MALA | $3000$ | $1.5 \cdot 10^{-2}$ | $4 \cdot 10^4$ |
| MLS | $6200$ | $2.7 \cdot 10^{-3}$ | $5.7 \cdot 10^4$ |

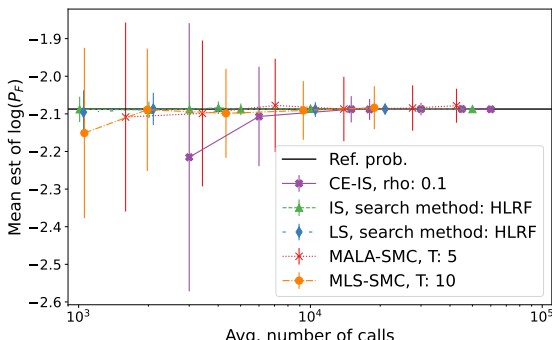

Figure 8: Convergence of different estimators w.r.t. the number of calls to the model $M_2$, on the input $\mathbf{x}_{0,3}$

Table 3: FORM/SORM estimations of $P_{\mathrm{F}} \approx 2.4 \cdot 10^{-7}$ for the custom CNN model, with uniform noise ($\varepsilon = 0.03$).

| Attack | $P_{\mathrm{F}}^{\mathrm{FORM}}$ | $P_{\mathrm{F}}^{\mathrm{SORM}}$ | $\cos(\tilde{u}^*, \nabla G(\tilde{u}^*))$ |
|---|---|---|---|
| CW | $3.91 \cdot 10^{-5}$ | NA | $-0.97$ |
| FMNA | $5.22 \cdot 10^{-5}$ | NA | $-0.985$ |
| HLRF | $2.16 \cdot 10^{-5}$ | NA | $-0.965$ |
| | $\|\tilde{u}^*\|_2$ | $G(\tilde{u}^*)$ | Time (in sec.) |
| CW | $3.95$ | $-1.2 \cdot 10^{-4}$ | $1.49$ |
| FMNA | $3.88$ | $-8.0 \cdot 10^{-5}$ | $0.23$ |
| HLRF | $4.09$ | $-8.1 \cdot 10^{-2}$ | $0.03$ |

sults, with the caveat that it systematically underestimates the probability of failure, which can be problematic when considering safety-critical applications. Focusing now on statistical estimators, we study their empirical convergence, for two images $\mathbf{x}_{0,1}$ and $\mathbf{x}_{0,2}$, with similar perturbations as in the previous section, i.e. uniform noise on $\ell_\infty$ balls of radius $\varepsilon = 0.18$. Simulation results are reported in Figure 8.

Like in previous experiments, the SMC-based algorithms converge much slower than both LS and the adversarial-attack-driven IS algorithm, though the gap is slightly less important in the case of input $\mathbf{x}_{0,3}$, which has a higher probability of failure, leading in particular to less dramatic underestimation of the MLS algorithm when using a smaller number of samples. Interestingly, in this example, the MLS algorithm, which is a black-box method, seems to slightly outperform the MALA-SMC algorithm that uses gradient information Tit et al. [2023].

## 5.3 CIFAR10

We move on to the CIFAR10 dataset, which is more challenging for rare event simulation as the dimension of each input is $d = 32^2 \times 3 = 3072$. We run experiments on a custom convolutional neural network, which contains four convolutional layers, followed by two dense layers and contains in total of $476\,278$ scalar parameters.

As before, we applied the FORM algorithm using different adversarial attacks, and the associated results are reported in Table 3. However, it is not possible to apply the SORM algorithm, as it requires too much memory capacity and computing power.

We next focus on the simulation algorithms' performance. Again, we primarily compare the LS and adversarial-attack-

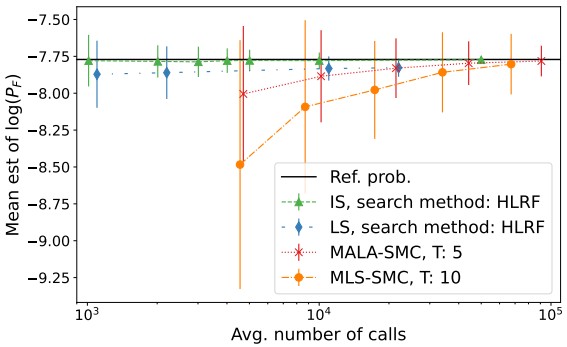

Figure 7: Convergence of the estimators w.r.t. the number of calls to the model $M_2$, on the input $\mathbf{x}_{0,2}$

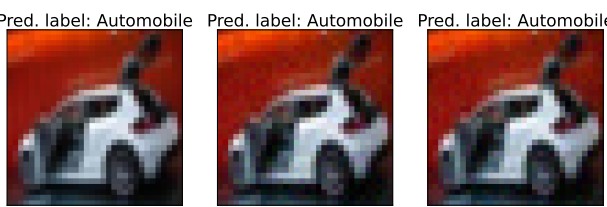

Figure 9: Clean input of the CIFAR10 dataset (on the left) and copies perturbed with Gaussian noise ($\sigma = 0.02$).

driven IS algorithm to sequential Monte Carlo methods used in the literature [Webb et al., 2019, Tit et al., 2023]. The associated results are reported in Figure 10 below.

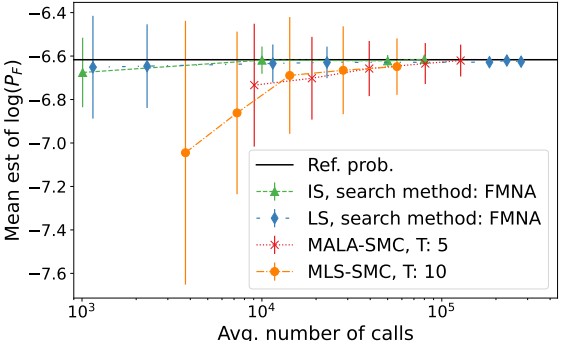

Figure 10: Convergence of different estimators w.r.t. the number of calls to the CNN.

We obtain similar results to that obtained for MNIST data: Our method and Line Sampling converge in a few thousand calls whereas state-of-the-art SMC algorithms require a few *hundreads* thousands of calls to obtain similar standard errors. That being said, the performance gap is somewhat smaller, a fact we attribute to the curse of dimension (COD), leading to weight degeneracy in Importance Sampling [Li et al., 2005].

Figure 11 compares the performance of the adversarial attacks. We notice again very small differences in terms of performance for the FMNA and HLRF algorithms. This means that the HLRF algorithm we have implemented for Neural Networks proves to be a powerful adversarial attack.

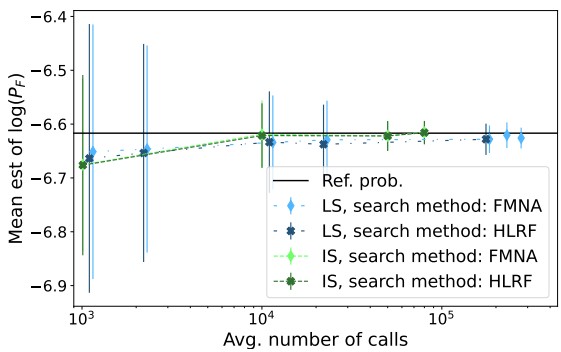

Figure 11: Convergence of different estimators w.r.t. the number of calls to the CNN.

### 5.4 IMAGENET RESULTS

Finally, we conclude this section with experimental results obtained on the ImageNet [Deng et al., 2009] dataset, where $d = 224^2 \times 3 = 150528$. We test the probabilistic robustness

of a pre-trained ResNet-18 model [He et al., 2015] under uniform noise of size $\varepsilon = 0.055$, around a clean image. Figure 12 illustrates the convergence of ADV-IS, MALA, and MLS estimation methods. In contrast to previous experiments, we see that the convergence rate of ADV-IS is worse than SMC-based methods. We attribute this poor performance to the high dimension of the problem, leading to catastrophic weight degeneracy, as mentioned above. In this case, it seems that SMC methods are more reliable than the proposed adversarial attack-based Importance Sampling. Thus, proposing a method that is both highly efficient for moderately high-dimensional data and reliable even for very high-dimensional data remains an important direction for future research in probabilistic robustness assessment.

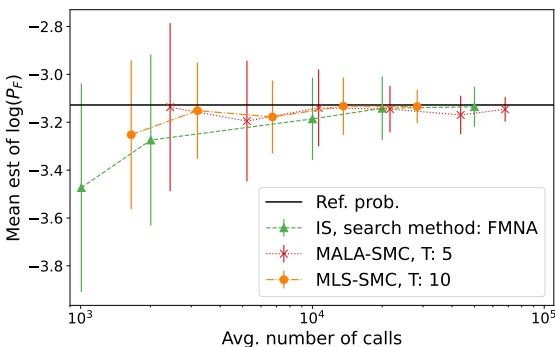

Figure 12: Convergence of different estimators w.r.t. the number of calls to the ResNet-18.

## 6 CONCLUSION

In conclusion, through extensive empirical analysis, we showed that the proposed algorithm outperforms, in terms of speed and computational efficiency, state-of-the-art methods for Neural Network reliability assessment, for moderately high dimensional datasets such as MNIST and CIFAR10. However, as mentioned above, a crucial limitation of our approach, compared to the sequential Monte Carlo approach, is the inability to handle very high-dimensional data. Indeed, while their algorithm is slower, Tit et al. [2023] show that it can efficiently estimate probabilities of failure on the ImageNet dataset. This limitation is directly linked to weight degeneracy, which becomes very difficult to handle when the problem dimension, $d$, is of the order of hundreds of thousands or more. Developing a hybrid approach between ours and splitting techniques, which has been done for another type of reliability problem [Jacquemart-Tomi et al., 2013], is a promising avenue for future research.

**Acknowledgements**

We thank French ANR and AID agencies for funding Chaire SAIDA ANR-20-CHIA-0011-01.

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
