# OpenReview forum: "Fast Reliability Estimation for Neural Networks with Adversarial Attack-Driven Importance Sampling"
_auai.org/UAI/2024/Conference — UAI 2024 poster_

### Official Review · Reviewer_BLwJ · 2024-03-18

**Q2-1 Originality-Novelty:** 1
**Q2-2 Correctness-Technical Quality:** 3
**Q2-5 Clarity Of Writing:** 4

**Q1 Summary And Contributions:**

This paper introduced a “reliability” (actually local robustness) assessment method combining Importance sampling and adversarial attacks, improving traditional Monte Carlo methods in terms of efficiency, while acknowledging its reliance on the effectiveness of adversarial attacks and its inability to handle very high-dimensional data such as ImageNet. Empirical experiments conducted over four models across three datasets, showing the claimed efficiency and accuracy.

**Q2-3 Extent To Which Claims Are Supported By Evidence:**

4: Excellent: all claims are supported by very convincing evidence (in the form of comprehensive experimental evaluation, rigorous mathematical proofs, detailed (pseudo-)code, precise references, well-motivated and realistic assumptions) and the authors deliver what they promise.

**Q2-4 Reproducibility:**

4: Excellent: key resources (e.g. proofs, code, data) are available and key details (e.g. proof sketches, experimental setup) are comprehensively described for competent researchers to confidently and easily reproduce the main results.

**Q3 Main Strengths:**

1.	The paper is carefully written, easy to follow.
2.	Technically sound solutions
3.	Sufficient experiments to validate the method.
4.	Good discussion on assumptions and limitations.

**Q4 Main Weakness:**

1.	Both the topic and the fundamental methods are well established. So, the novelty is limited by combining them.
2.	The term “reliability” used is confusing.

**Q5 Detailed Comments To The Authors:**

In the AI/ML community, it is called “probabilistic local robustness” as those papers cited in the paper. I guess, by terming the problem as a reliability problem, the authors would like give credit to the Statistical Reliability Engineering community. But reliability is normally defined as, in international standards like IEC 61508, the ability of a *product, service, or system* to perform its required functions under stated conditions for a specified period of time. So, it should be defined at the NN-level considering all inputs, rather than local-instance level concerning only one given input (Cf. Fig.1 in [1]). Although the mathematical solution is the same, application examples in those “Statistical Reliability Engineering” papers would be also concerning the whole product rather than a single input. Also, software reliability is a user-centric concept [2], it really depends on how users operate the software (inc. AI software)—the operational profile (OP) [3]. The local perturbation distribution modelled in this work is not a OP, so again, it is confusing to name it reliability.
That said, I still like the paper and strongly suggest the author to change reliability to local robustness, to lower the language barriers between different communities.

A minor: is \phi in eqn (4) defined?

[1] Dong, Y., Huang, W., Bharti, V., Cox, V., Banks, A., Wang, S., ... & Huang, X. (2023). Reliability assessment and safety arguments for machine learning components in system assurance. ACM Transactions on Embedded Computing Systems, 22(3), 1-48.

[2] Littlewood, B., & Strigini, L. (2000, May). Software reliability and dependability: a roadmap. In Proceedings of the Conference on the Future of Software Engineering (pp. 175-188).

[3] Musa, J. D. (1993). Operational profiles in software-reliability engineering. IEEE software, 10(2), 14-32.

**Q9 Complying With Reviewing Instructions:**

Yes

---

> ### Author Rebuttal · Authors · 2024-04-05
>
> We gratefully acknowledge your appraisal of our work as well as your constructive criticism of our work.
>
> 1. “Both the topic and the fundamental methods are well established. So, the novelty is limited by combining them”: This is true, both Importance Sampling and Adversarial Attacks are well-established subjects. Since the connection had not been made we still consider this a worthy endeavor.
>
> 2. “The term “reliability” used is confusing”: We understand your concern and thus, we will replace in the text the term “reliability” with the term “probabilistic local robustness”, which is more precise. To be fair we used the term “reliability” in the sense it is used in the field of structural reliability, where the assessed model is static rather than dynamic, and which aligns quite closely with the problem at hand. However, you made a very good point that this term in software (including machine learning) has a quite different meaning and we will thus replace it, which will make the paper more readable.
>
> About $\phi$ in equation (4): you are absolutely right it was not defined, and this notation also appears in equations (14). We use $\phi$ to denote the probability density function of the standard normal distribution, and similarly, we use $\Phi$ to denote the cumulative distribution function of the same distribution. Thank you for this comment, we will add definitions of these important notations.

---

### Official Review · Reviewer_KYEL · 2024-03-23

**Q2-1 Originality-Novelty:** 3
**Q2-2 Correctness-Technical Quality:** 3
**Q2-5 Clarity Of Writing:** 3

**Q1 Summary And Contributions:**

The paper proposes a new method of assessing neural network reliability, using importance sampling and adversarial examples. The authors show empirically that the proposed method is faster than existing alternatives.

**Q2-3 Extent To Which Claims Are Supported By Evidence:**

3: Good: the main claims are supported by convincing evidence (in the form of adequate experimental evaluation, proofs, (pseudo-)code, references, assumptions).

**Q2-4 Reproducibility:**

2: Fair: key resources (e.g. proofs, code, data) are unavailable but key details (e.g. proof sketches, experimental setup) are sufficiently well-described for an expert to confidently reproduce the main results.

**Q3 Main Strengths:**

The method seems novel, although I am not close to the topic, and I may miss some works in my brief literature review. The evaluation sounds legitimate since the author justified that high-dimensional data cannot be used.

**Q4 Main Weakness:**

All the evaluations are done over the image data. Since the method is tailored to the low-dimensional data, tabular data might be more suitable for such evaluation.

Assumptions stated in Section 4.2 are not tied to some real-world constraints, and therefore, it is hard for a reader to judge beforehand if they hold or not.

**Q5 Detailed Comments To The Authors:**

1) Short timing observations are reported in Table 1, without confidence intervals (0.01 sec)

**Q9 Complying With Reviewing Instructions:**

Yes

---

> ### Author Rebuttal · Authors · 2024-04-05
>
> Thank you for reviewing our paper. Your comments will help improve this work.
>
> 1. “All the evaluations are done over the image data. Since the method is tailored to the low-dimensional data, tabular data might be more suitable for such evaluation “:  Following your comment and that of reviewer 7V4G, we will add experiment results on a tabular dataset.
>
> 1bis : "Short timing observations are reported in Table 1, without confidence intervals (0.01 sec)": You are right.  We computed empirical standard deviations for these metrics but we forgot to report them. We will report them in the final text.
>
> 2. “Assumptions stated in Section 4.2 are not tied to some real-world constraints, and therefore, it is hard for a reader to judge beforehand if they hold or not “: It is true that this properties can hardly be verified before hand. However, we do provide a diagnostic the second assumation (A2) quantity which always be computed a posteriori.

---

### Official Review · Reviewer_7V4G · 2024-03-23

**Q2-1 Originality-Novelty:** 3
**Q2-2 Correctness-Technical Quality:** 3
**Q2-5 Clarity Of Writing:** 3

**Q1 Summary And Contributions:**

This paper introduces a novel approach to evaluate the reliability of Neural Networks (NNs) by integrating adversarial attacks with Importance Sampling (IS), enhancing the assessment’s precision and efficiency. The  research not only presents an innovative strategy for reliability assessment in NNs but also sets the stage for further work exploiting the connection between adversarial robustness and the field of statistical reliability engineering.

**Q2-3 Extent To Which Claims Are Supported By Evidence:**

2: Fair: the main claims are somewhat supported by evidence (but the experimental evaluation may be weak, or does not match entirely with the claims, important baselines may be missing, proofs contain important ideas but lack rigor, algorithmic details are only discussed superficially, references are imprecise, assumptions are not sufficiently motivated or explicated, etc.).

**Q2-4 Reproducibility:**

2: Fair: key resources (e.g. proofs, code, data) are unavailable but key details (e.g. proof sketches, experimental setup) are sufficiently well-described for an expert to confidently reproduce the main results.

**Q3 Main Strengths:**

1. The authors introduce a novel sample method called improtance sampling and provide the proof and math formula.
2. The IS method significantly improve the speed of converge.

**Q4 Main Weakness:**

1. The authors only provide the experiments on the simple networks like MLP and CNN.
2. The authors only conduct experiments on MNIST and CIFAR10 dataset. It would be better to conduct more experiments to demonstrate the effectiveness of the method.

**Q5 Detailed Comments To The Authors:**

see weakness.

**Q9 Complying With Reviewing Instructions:**

Yes

---

> ### Author Rebuttal · Authors · 2024-04-05
>
> We gratefully acknowledge your review of our work. We address some concerns below:
>
> 1. "The authors only provide the experiments on the simple networks like MLP and CNN. ":  We will endeavor to add results on a transformer trained for classification on the IMDB reviews dataset.
>
> 2. "The authors only conduct experiments on MNIST and CIFAR10 dataset. It would be better to conduct more experiments to demonstrate the effectiveness of the method" : Following your comment and that of reviewer KYEL, we will add experiment results on ImageNet on which we have already experimented and endeavor to provide results on a tabular dataset.

---

### Official Review · Reviewer_w2Gg · 2024-03-24

**Q2-1 Originality-Novelty:** 4
**Q2-2 Correctness-Technical Quality:** 4
**Q2-5 Clarity Of Writing:** 3

**Q1 Summary And Contributions:**

In this work, the authors introduce a method for estimating the reliability of Neural Networks (NN) by combining adversarial attacks with Importance Sampling (IS), aiming to improve the precision and efficiency of the assessment. This novel approach identifies vulnerable input regions, offering a directed alternative to traditional Monte Carlo methods. It compares favorably against classical reliability techniques and simulation methods in empirical validations using MNIST and CIFAR10 datasets, demonstrating its potential for accurate NN reliability estimation despite limitations like dependency on adversarial attack effectiveness and challenges with high-dimensional data.

**Q2-3 Extent To Which Claims Are Supported By Evidence:**

3: Good: the main claims are supported by convincing evidence (in the form of adequate experimental evaluation, proofs, (pseudo-)code, references, assumptions).

**Q2-4 Reproducibility:**

3: Good: key resources (e.g. proofs, code, data) are available and key details (e.g. proofs, experimental setup) are sufficiently well-described for competent researchers to confidently reproduce the main results.

**Q3 Main Strengths:**

1. It is novel and interesting about the integration of adversarial attacks with Importance Sampling to guide the sampling process toward vulnerable regions of the input space is a novel approach.

2. It has been empirically validated on two well-known datasets, MNIST and CIFAR10, across a variety of models, which demonstrate the capability of the approach to accurately estimate NN reliability.

**Q4 Main Weakness:**

1. The effectiveness of the proposed method is inherently tied to the efficiency of adversarial attacks. If the adversarial attacks are not effective, the reliability estimation may not be accurate. Also, it is encouraged to adopt more recent attacks to validate the effectiveness.

2. The paper acknowledges a limitation in handling very high-dimensional data, such as ImageNet, due to weight degeneracy. This restricts the applicability of the method to datasets of moderate dimensionality. I would appreciate it if the authors could validate the effectiveness of such a challenging dataset.

3. I am also worried about the computational cost using such method.

4. I suggest the authors detailedly introduce the relationship between the magnitude of estimation results and robustness.

**Q5 Detailed Comments To The Authors:**

See weakness

**Q9 Complying With Reviewing Instructions:**

Yes

---

> ### Author Rebuttal · Authors · 2024-04-05
>
> We thank you for your comments and try to answer some of your concerns below.
>
> 1. “The effectiveness of the proposed method is inherently tied to the efficiency of adversarial attacks [...] Also, it is encouraged to adopt more recent attack “ :  We understand the reviewer’s concern about the dependency of the proposed approach on the efficiency of adversarial attacks, which we tried to state clearly in the article. We would however like to make 2 additional comments about this. First, like the vanilla Monte Carlo method, our is always consistent and thus will eventually converge to the true probability of failure given enough computing power. We will recall this consistency property in the text. Secondly, we do have a convergence diagnostic for adversarial attacks we can help detect cases where a more sophisticated method is needed. One fact we have not mentioned is that in general the near-optimal shift parameter \theta^* found using Cross-Entropy based IS is actually aligned with the adversarial attack, which we verify by computing the cosinus similarity between these vectors, which is very close to 1. We will give more details about this in a final version. In the cases where the adversarial are found not to converge, one can still use them as a warm start (i.e. as a \theta_0) for CE-IS. We have used the best white-box adversarial attack we know of considering that they should be especially good at find adversarial attacks of near-minimum norm as fast as possible. If the reviewer knows of more recent white-box attacks performing better in that specific aspect would welcome it.
>
> 2. “The paper acknowledges a limitation in handling very high-dimensional data, such as ImageNet” | “I would appreciate it if the authors could validate the effectiveness of such a challenging dataset.” : We will add the experimental results obtained on ImageNet.
>
> 3. “I am also worried about the computational cost of using such method.” : the computation cost of our method is relatively low and is the main comparative advantage of our method, as illustrated in the experiments section, as only a few thousand calls are needed to obtain a precise estimation of even low failure probability vs. tens to hundreds thousands of calls needed for the previous state-of-the-art.
>
> 4. “I suggest the authors detailedly introduce the relationship between the magnitude of estimation results and robustness.” : we assume you mean the relationship between the probability of failure and robustness. The main idea is that the lower this probability, the more the prediction is robust to input noise. We will try to make it clearer.

---

### Meta-Review · Area_Chair_ecwi · 2024-04-16

The paper introduces a method for estimating the reliability of Neural Networks by combining adversarial attacks with Importance Sampling, aiming to improve the precision and efficiency of the assessment. All the reviewers support the publication of the paper. Please add the discussions from the rebuttals and revise the manuscript for the final version.